# Asymmetric opening of the homopentameric 5-HT$_{3A}$ serotonin receptor in lipid bilayers

Yingyi Zhang [1,2,7,11], Patricia M. Dijkman [1,2,8,9,10,11], Rongfeng Zou[3], Martina Zandl-Lang[4], Ricardo M. Sanchez[1,2], Luise Eckhardt-Strelau[1], Harald Köfeler [5], Horst Vogel[3,6], Shuguang Yuan[3✉] & Mikhail Kudryashev [1,2✉]

Pentameric ligand-gated ion channels (pLGICs) of the Cys-loop receptor family are key players in fast signal transduction throughout the nervous system. They have been shown to be modulated by the lipid environment, however the underlying mechanism is not well understood. We report three structures of the Cys-loop 5-HT$_{3A}$ serotonin receptor (5HT$_3$R) reconstituted into saposin-based lipid bilayer discs: a symmetric and an asymmetric apo state, and an asymmetric agonist-bound state. In comparison to previously published 5HT$_3$R conformations in detergent, the lipid bilayer stabilises the receptor in a more tightly packed, 'coupled' state, involving a cluster of highly conserved residues. In consequence, the agonist-bound receptor conformation adopts a wide-open pore capable of conducting sodium ions in unbiased molecular dynamics (MD) simulations. Taken together, we provide a structural basis for the modulation of 5HT$_3$R by the membrane environment, and a model for asymmetric activation of the receptor.

[1] Max Planck Institute of Biophysics, Frankfurt am Main, Germany. [2] Buchmann Institute for Molecular Life Sciences (BMLS), Goethe University of Frankfurt, Frankfurt am Main, Germany. [3] Research Center for Computer-Aided Drug Discovery, Shenzhen Institute of Advanced Technology, Chinese Academy of Sciences, Shenzhen, China. [4] Division of General Pediatrics, Department of Pediatrics and Adolescent Medicine, Medical University of Graz, Graz, Austria. [5] Core Facility Mass Spectrometry, ZMF, Medical University of Graz, Graz, Austria. [6] Institute of Chemical Sciences and Engineering (ISIC), Ecole Polytechnique Fédérale de Lausanne (EPFL), Lausanne, Switzerland. [7] Present address: Biological Cryo-EM Center, Hong Kong University of Science and Technology, Clear Water Bay, Kowloon, Hong Kong, China. [8] Present address: Max Planck Institute of Biochemistry, Martinsried, Germany. [9] Present address: Institute of Neuropathology, University Medical Center, Göttingen, Germany. [10] Present address: Cluster of Excellence "Multiscale Bioimaging: from Molecular Machines to Networks of Excitable Cells" (MBExC), University of Göttingen, Göttingen, Germany. [11] These authors contributed equally: Yingyi Zhang, Patricia M. Dijkman. ✉email: shuguang.yuan@siat.ac.cn; misha.kudryashev@biophys.mpg.de

Synaptic pLGICs play a central role in fast neuronal signalling, translating neurotransmitter release into an electrical signal by opening an ion channel gate depolarising the transmembrane potential[1,2]. From recent high-resolution structures of various pLGICs, a general model for the function of these receptors has evolved[2]. Five protein subunits are arranged in homo- or heteropentameric pseudo-symmetry around a central ion-conducting pore. Ligand binding at the subunit interfaces at the extracellular domain (ECD) is relayed to the 6-nm-distant transmembrane domain (TMD) comprised of four α-helices per subunit (M1–M4). For some pLGICs, M4 extends into the MA helix, forming an intracellular helical bundle, which together with a disordered region makes up the intracellular domain (ICD).

The membrane environment has been shown to modulate key aspects of neurotransmission through pLGICs, including agonist binding, gating, and receptor conformation[3–6]. Early studies on nicotinic acetylcholine receptors (nAChRs) a mixture of unsaturated phosphatidylethanolamine (PE) and phosphatidic acid (PA) or phosphatidylserine (PS) lipids in combination with cholesterol best supports native-like function in reconstituted systems[3–6]. Interactions between the extracellular end of M4 of the nAChR and cholesterol, in particular, have been suggested to be of importance, with the receptor adopting an 'uncoupled' state refractory to agonist activation in the absence of cholesterol[7]. Similarly, previous reports showed diminished serotonin-induced currents upon cholesterol depletion in cells expressing 5HT₃R[8]. Lipid–protein interactions have since been observed in a number of structural studies of other pLGICs[9–12], where they were proposed to induce pore expansion and potentiate agonist binding[11], have a role in trafficking[9], or provide the rigidity necessary for productive gating[12,13]. However, despite the clear role that lipids play in modulating pLGIC function, the underlying molecular mechanism is still not well understood.

Here, we employ single-particle cryo-electron microscopy (cryo-EM) in combination with MD simulations to elucidate the gating mechanism of the homopentameric 5HT₃R and its modulation by lipids. Besides being a therapeutic target[14], the 5HT₃R represents an interesting case as despite several structures having been reported for the receptor in detergent in the presence and absence of agonist[15–18], the mechanism of gating remains an open question[19]. This is in part due to difficulties in unambiguously assigning functional states to static structures and the notable differences observed between the structures of the serotonin-bound receptor in detergent micelles[17,18]. We report the structures of 5HT₃R reconstituted into lipid bilayers in the presence and absence of its native agonist serotonin (5-HT). Compared to the previously published structures of detergent-solubilised 5HT₃R[16–18], the lipid-embedded receptor displays a more tightly packed, 'coupled' state. Moreover, the receptor TMD is considerably compressed along the central axis due to the thickness of the hydrophobic core of the lipid bilayer. In consequence, in lipid bilayers, the serotonin-bound receptor can adopt a substantially wider and asymmetric pore as compared to the detergent-solubilised form, allowing free diffusion of sodium ions across the membrane in unbiased MD simulations. In addition to a symmetric resting state, the lipid-embedded receptor adopted an asymmetric conformation in the absence of ligand. Based on MD simulations we hypothesise that the asymmetric apo conformation represents an intermediate activation state. Taken together, our results provide a structural basis for the modulation of 5HT₃R activity by the lipid membrane environment and broaden our understanding of 5HT₃R gating.

## Results

**Cryo-EM of lipid bilayer-embedded 5HT₃R reveals symmetric and asymmetric structures.** We reconstituted the full-length murine 5HT₃R into saposin-based lipid bilayer discs (5HT₃R-Salipro, Supplementary Fig. 1a, b)[20] composed of brain polar lipid extract as verified by lipidomic analysis (Supplementary Table 1). The affinity of 5HT₃R-Salipro for serotonin (with a dissociation constant of 300 ± 90 and 80 ± 40 nM in the absence and presence of 2 mM CaCl₂, respectively, Supplementary Fig. 1c), was consistent with literature values for membrane-embedded 5HT₃R[21,22]. The apparent melting temperature of the receptor was higher for 5HT₃R-Salipro than for the detergent-solubilised receptor ($T_m = 66.2 \pm 0.2\,°C$ vs $61.5 \pm 0.3\,°C$, Supplementary Fig. 1d), suggesting that the lipid bilayer stabilises the receptor, in agreement with previous observations[23].

Using single-particle cryo-EM, we solved structures of 5HT₃R in the absence (apo-5HT₃R-Salipro) and presence (serotonin-5HT₃R-Salipro) of serotonin in apparent closed- and open-pore conformations, respectively (Fig. 1a, b and Supplementary Fig. 1e, f). The final apo-5HT₃R-Salipro particle set yielded a main C5 symmetric map (apo-C5, from 61% of the particles), as well as an asymmetric map (apo-C1, from 39% of the particles) at nominal resolutions of 3.2 and 3.1 Å, respectively (Supplementary Figs. 2–4). In contrast, the serotonin-5HT₃R-Salipro data set yielded only an asymmetric map at a nominal resolution of 2.8 Å (Supplementary Figs. 2–4). For all maps, local resolution varied and was highest at the ECD and lowest at the ICD (Supplementary Figs. 3d and 4).

To assess the degree of (a)symmetry of both apo maps, we performed symmetry expansion[24] of each particle set; each pentamer was copied and rotated five times around its C5 (pseudo)symmetry axis, followed by signal subtraction to retain one copy of each monomer, and subsequent 3D classification without alignment. While no notable differences were found between the individual monomer classes obtained from the symmetry expanded apo-C5 particle set (at the ~4.3–4.6 Å resolution estimated by 3D classification in RELION), clear differences were found in the case of the apo-C1 particle set (Supplementary Fig. 5a–c). At the obtained resolution, all monomer classes could be divided into five different monomer groups (A–E) representing the five monomers resolved in the apo-C1 consensus map (Supplementary Fig. 5a–c). The different monomer types were ranked from most to least deviating from the apo-C5 symmetric pentamer arrangement as A < E < C < D < B based on backbone root-mean-square deviation (RMSD, Supplementary Fig. 5b). Interestingly, the percentage of monomer particles corresponding to each group follows the same order, with 'A' type monomers being observed most often (Supplementary Fig. 5a). To assess whether the asymmetric apo receptor was composed of a random mixture of the resolved monomer classes or adopted preferential arrangements, we subsequently investigated the subunit composition of the corresponding pentamers in the apo-C1 data set. Nearly all possible different combinations were observed. The most prevalent configuration corresponded to the resolved consensus structure (12% of pentamers, Supplementary Fig. 5b, d), with the rest of the top 5 (31% of pentamers in total) corresponding to the same configuration but with one or two monomers swapped for an 'A' type monomer, i.e. a monomer deviating less from the symmetric apo-C5 structure. Scoring all pentamers based on their deviation from the apo-C5 structure resulted in a distribution skewed towards more symmetric pentamers (Supplementary Fig. 5e). Thus, these results suggest that while the apo/resting state is dynamic, configurations deviating less from symmetry are more energetically favourable. This is also in line with the overall particle distribution of the apo

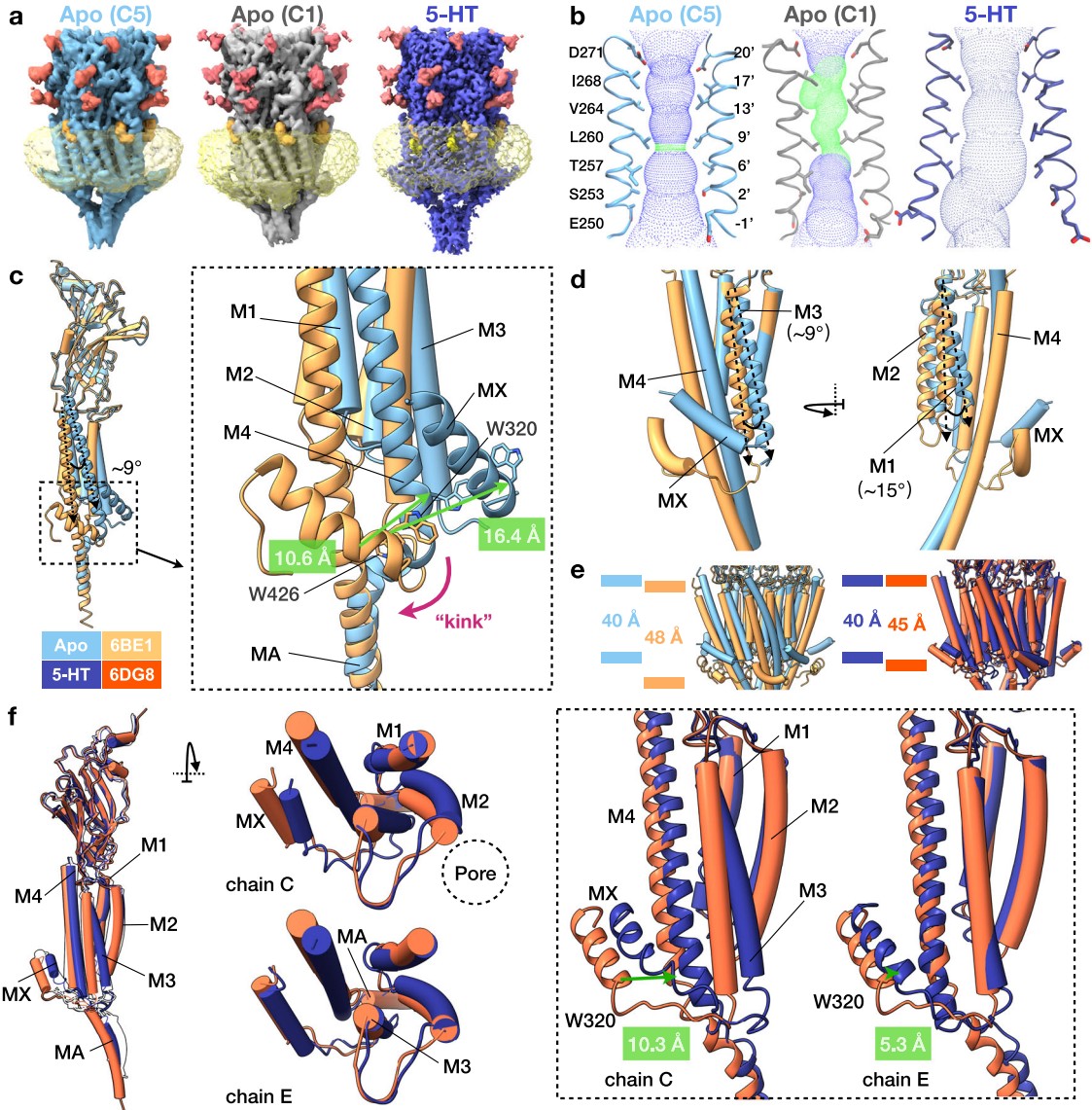

**Fig. 1 Global architecture of 5HT₃R-Salipro. a** Cryo-EM maps of apo- (C5 symmetric, light blue; C1 symmetric, grey) and serotonin-5HT₃R-Salipro (5-HT, dark blue), with red, orange, yellow and transparent-yellow densities corresponding to glycosylation, cholesterol, inter-subunit phospholipids and the saposin-lipid bilayer, respectively. **b** Static pore radius of the 5HT₃R-Salipro models determined using HOLE[64]. Green and blue spheres represent radii of 1.8–3.3 Å, and >3.3 Å, respectively. **c, d** Superposition of the apo-5HT₃R-Salipro (C5) and apo-5HT₃R-detergent (6BE1, yellow) structures (single subunit is shown for clarity). Differences at the **c** MX and M4-MA helices, and **d** M1 and M3 helices are highlighted. **e** The TMD thickness was measured between residues L221 and W426 for the 5HT₃R-Salipro and the 5HT₃R-detergent structures (coloured as in **c** and **f**). **f** Superposition of the monomers of the serotonin-5HT₃R-Salipro and serotonin-5HT₃R-detergent (6DG8, orange) structures. The left panel shows chain C in colour and black silhouettes for the other chains. The middle panel shows a top view of the TMD. Right panel highlights differences in the position of MX (8.2, 6.7, 10.3, 7.0 and 5.3 Å measured at the Cα atom of W320 of chains A–E, respectively). Chains C and E represent the most and least different subunits, respectively.

data set, with 61% of particles falling into the C5 symmetric class and 39% into the asymmetric class. Furthermore, the symmetric and asymmetric apo states showed globally similar conformations, that differed mostly in the relative positioning of the TMDs of the different subunits (Supplementary Table 2, and discussed below). Thus, unless stated otherwise, the main symmetric apo conformation (referred to as apo-5HT₃R-Salipro) is used in subsequent analyses.

Analogous symmetry expansion analysis of the serotonin-5HT₃R-Salipro particle set also revealed residual heterogeneity (discussed in more detail below), but no further high-resolution reconstructions could be obtained. Thus, the ligand-bound receptor in lipid bilayers is also dynamic, and the resolved serotonin-5HT₃R-Salipro structure represents a consensus structure.

**Global architecture of 5HT₃R-Salipro structures.** Both the apo- and serotonin-5HT₃R-Salipro conformations differ substantially from the previously reported detergent-solubilised structures (Fig. 1c–f and Supplementary Movie 1)[15–18]. To assess the effect of lipids on the receptor, the cryo-EM-derived models with PDB accession codes 6BE1 and 6DG8 are used as references for the apo and serotonin-bound states in detergent (apo-5HT₃R-detergent and serotonin-5HT₃R-detergent), respectively, as they were solved in the absence of any exogenously added lipids[16,18]. In the apo state, the lipid-facing M1, M3, and M4 helices of 5HT₃R-Salipro are significantly tilted compared to their positions in detergent. The largest displacement is observed for M4 which is tilted by ~9°, displacing W426 on the intracellular side by 10.6 Å (Fig. 1c). The MX helices were shifted by 16.4 Å (measured

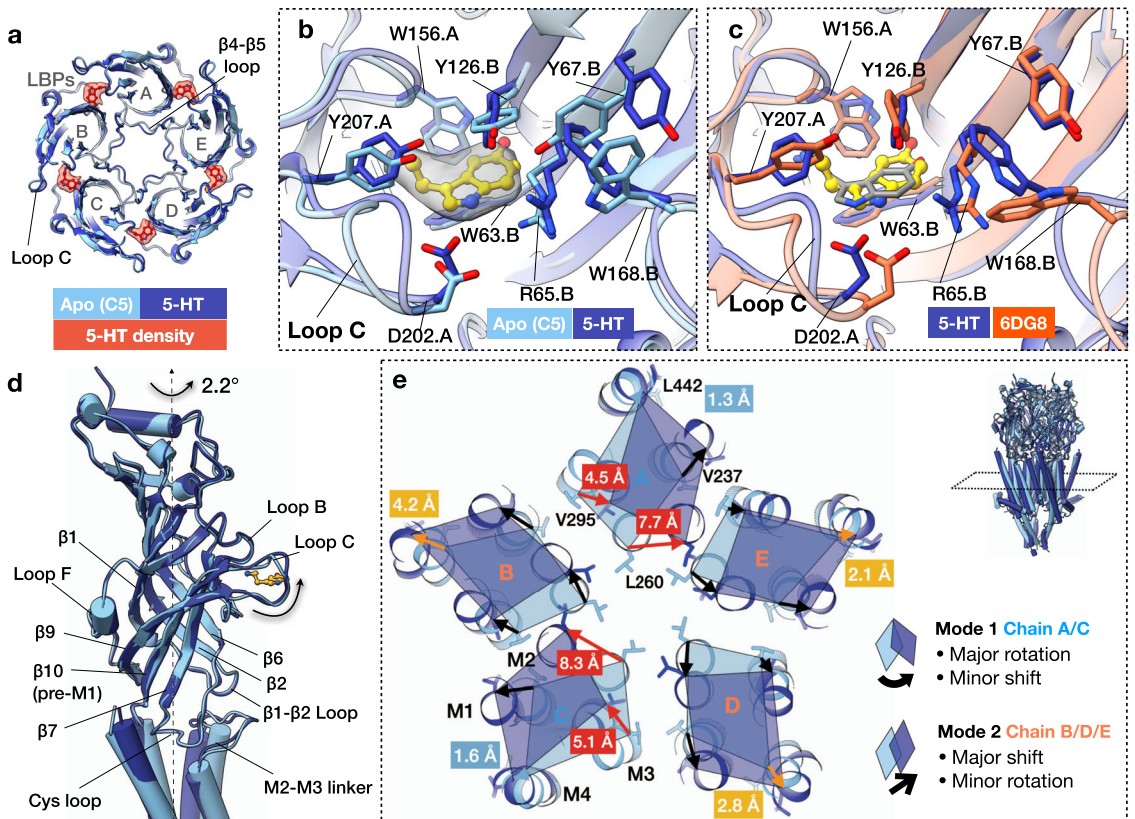

**Fig. 2 Serotonin-induced conformational changes of 5HT₃R-Salipro. a** Superposition of the ECD of the apo- and serotonin-5HT₃R-Salipro structures, showing a cross-section at LBPs. Five densities (transparent red, contoured at σ 8.0) corresponding to serotonin (5-HT) were observed in the serotonin-5HT₃R-Salipro map. **b** Close-up of the LBP showing serotonin (yellow) and the corresponding density (transparent grey, contoured at σ 8.0). Residues on the primary and complementary subunits are labelled '.A' and '.B', respectively. **c** LBPs of the serotonin-bound 5HT₃R-Salipro and 5HT₃R-detergent (6DG8) structures. Serotonin molecule of 6DG8 is displayed in grey. **d** Superposition of the ECD of the apo- and serotonin-5HT₃R-Salipro structures, showing the ECD and ECD/TMD interface. **e** Superposition of the apo- and serotonin-5HT₃R-Salipro states showing a cross-section at the TMD residue L260 (9' position) of M2. Displacements measured at the Cα atoms of the indicated residues on each helix in the same cross-section are shown (see also Supplementary Table 3). Two modes of rearrangements are observed: (1) major rotation/minor shift for chains A/C, and (2) minor rotation/major shift for chains B/D/E.

at W320, Fig. 1c). Helices M1 and M3 are likewise tilted by ~15° and ~9° (Fig. 1d), respectively, yielding displacements of ~7 Å (measured at the intracellular residues C243 and L307, respectively). Consequently, the receptor in Salipro is markedly shorter along the central channel axis than in detergent (40 vs 48 Å at the TMD, Fig. 1e, see also Supplementary Movie 1). These rearrangements place W426 and W320 at the energetically favourable headgroup region of the lipid bilayer[25]. The compression of the TMD along the pore axis compared to the 5HT₃R-detergent structures is likely a direct consequence of the thickness of the hydrophobic core of the lipid membrane which is less malleable than a detergent micelle surrounding the receptor. The resulting TMD thickness is in line with reports of other pLGICs in nanodiscs (~38–39 Å for the GABAₐ receptor (GABAₐR)[9,26] and ~43 Å for nAChR[27]). Comparable, albeit slightly more modest, differences were observed between the detergent and Salipro serotonin-bound conformations (Fig. 1e, f).

**Serotonin-induced conformational changes.** Density corresponding to serotonin was resolved in all five ligand-binding pockets (LBPs) of the serotonin-5HT₃R-Salipro map (Fig. 2a). No significant differences were observed between the ECDs of the five subunits (RMSD ~0.7–0.8 Å, Supplementary Table 2) at the resolution obtained (~3 Å at ECD domain). Ligand binding resulted in the inward displacement of loop C (capping) compared to the apo state, as previously reported for other pLGICs[28],

stabilised by an electrostatic interaction (~5 Å) between D202 (loop C) on the principal subunit and R65 (β2) on the complementary subunit (Fig. 2b). Concomitantly, W168 (post-β8) and Y67 (β2) on the complementary subunit adopt different rotamer positions stabilised by a cation-π interaction between R65 and W168 and π–π stacking between W168 and Y67, which was not observed in the serotonin-5HT₃R-detergent form where loop C capping was less pronounced and the LBP was overall less compact (Fig. 2c). Ligand binding induces a 2.2° counterclockwise rotation of the entire ECD around its own axis as viewed from the extracellular side, repositioning the loops at the ECD-TMD interface (Fig. 2d): the β1–β2 loop (via β2), the Cys loop (via loop B, β6, and β7), loop F (via loop C capping transmitted through β9, and via the rotamer switch of W168 on the complementary side), and pre-M1 (via loop C capping transmitted through β10). As a result, the M2–M3 linker of the principal subunit is repositioned by 2.5–2.9 Å (measured at L273) relative to the apo-5HT₃R-Salipro form (Fig. 2d), relaying the conformational changes of the ECD to the TMD.

At the TMD, ligand-binding results in a clockwise rotation and translation for each subunit (viewed from the ECD along the pore axis), resulting in a larger pore diameter (Fig. 2e). Based on its pore profile, serotonin-5HT₃R-Salipro represents a more open conformation than the most open serotonin-bound state previously solved in detergent (6DG8, serotonin-5HT₃R-detergent), with a minimum pore diameter of 9.4 Å (at residue I267) vs 6.7 Å (at residue L260, Supplementary Fig. 1e). The asymmetry of

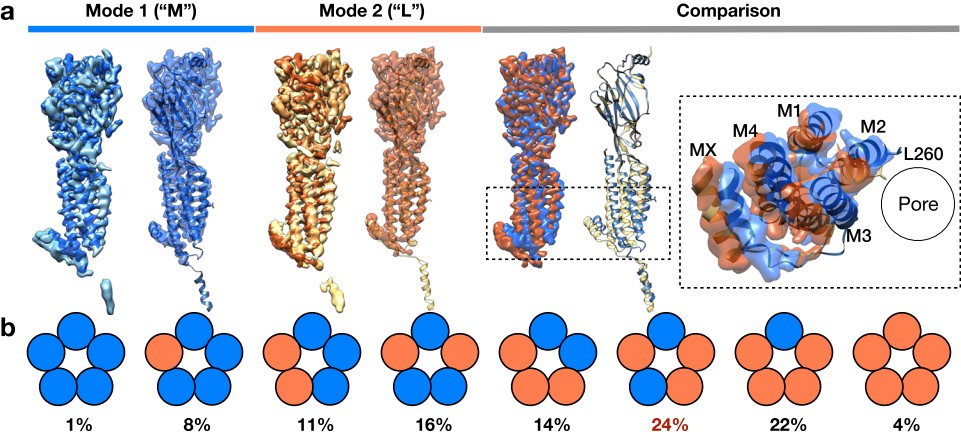

**Fig. 3 Conformational plasticity of serotonin-5HT₃R-Salipro state. a** Classes resulting from symmetry expanded monomer-focused 3D classification were divided into two groups based on comparison to the serotonin-5HT₃R-Salipro model: (1) densities similar to chain A/C (corresponding to 'Mode 1' in Fig. 2e) or M-type (left, model of chain C is shown), and (2) densities similar to chain B/D/E (Mode 2), or L-type (middle, model of chain D is shown). Superposition of M- and L-types with insert showing a top view (right). **b** Schematic showing the population of the eight different combinations of serotonin-5HT₃R-Salipro arrangements; blue and orange circles represent M- and L-type subunits, respectively. The largest class corresponds to the consensus serotonin-bound conformation.

serotonin-5HT₃R-Salipro was most pronounced at the TMD, with different conformations observed for the individual subunits. Taking a cross-section through the middle of the TMD at L260, two modes of rearrangements could be identified (Fig. 2e): (1) a major rotation and minor shift (chain A and C); and (2) minor rotation, but major shift (chain B, D and E). The RMSD between subunits within each of the two groups is ~2 Å, while between groups it is ~3–4 Å (Supplementary Table 2). Mode 1 resulted in the largest displacement (7.7–8.3 Å at L260, and 4.5–5.1 Å at V295 (M3)) relative to apo-5HT₃R-Salipro, yielding a 'more-open' conformation (Fig. 2e and Supplementary Table 3). Overall smaller displacements relative to apo-5HT₃R-Salipro, though still larger than observed for serotonin-5HT₃R-detergent, were found for mode 2 (4.7–5.1 at L260 and 2.7–3.9 at V295), yielding a 'less-open' conformation.

**Residual conformational plasticity of the asymmetric open state**. As already stated, to investigate any residual heterogeneity in the serotonin-5HT₃R-Salipro particle set, we performed symmetry expansion of the pentamers and 3D classification of the monomers (Fig. 3a). The resulting monomer classes all represented open-state conformations with clear density for serotonin. Based on comparison to the different subunits of the serotonin-5HT₃R-Salipro model, the classes were sorted into two groups: 'more' (M), and 'less' (L) open conformations, corresponding to modes 1 and 2, respectively (Fig. 2e). Using this assignment, the corresponding pentamers were grouped based on their composition of subunits in different states (M or L, Fig. 3b). The largest group of particles (24%) fell into class 'LMLML' corresponding to a pentamer containing two non-consecutive 'more' open subunits, in agreement with the resolved consensus serotonin-5HT₃R-Salipro map. All other possible combinations were observed. Thus, we propose that a conformational spectrum of asymmetric open states exists, while the resolved conformation represents the energetically most favourable form under our experimental conditions. Interestingly, classes containing non-consecutive 'M'-subunits were more commonly found than classes containing the same number of 'M'-subunits in a consecutive arrangement, paralleling observations from single-channel recordings that non-consecutive ligand binding promotes receptor activation[29].

**Lipids support pore opening in 5HT₃R**. To investigate the ion permeation pathway in more detail, we performed triplicate unbiased 200-ns all-atom MD simulations in an explicit lipid bilayer for the apo-5HT₃R-Salipro and serotonin-5HT₃R-Salipro models, as well as the serotonin-5HT₃R-detergent model for comparison. Excluding the MX helices, the simulation systems stabilised within ~100 ns, and we used the last 50 ns (150–200 ns) for analysis (Supplementary Fig. 6). The MX helices are partially disordered and poorly resolved in our EM maps; as in previous simulations of the 5HT₃R[19], they were highly flexible during the simulations as the link between them and the ICD is fully disordered, and not included in the model.

Significant differences in the conformation of M2 were observed between the models during the simulations. The M2 helices adopted an 'A' shape with respect to the central pore axis in the serotonin-5HT₃R-Salipro form, while in the apo-5HT₃R-Salipro and serotonin-5HT₃R-detergent forms the helices adopted an 'H' and 'V' shape, respectively (Supplementary Fig. 7a). Concomitantly, M2 was observed to kink in different positions in the serotonin-5HT₃R-Salipro and serotonin-5HT₃R-detergent forms, with M2 in the former mainly bending at the intracellular residue V252, and in the latter at the extracellular residue I268 (Supplementary Fig. 7b). Furthermore, the root-mean-square fluctuation of M2 indicated that the serotonin-5HT₃R-Salipro form is much more flexible than both the apo-5HT₃R-Salipro and serotonin-5HT₃R-detergent forms (Supplementary Fig. 7c).

No water molecules were observed in the central ion pore region during simulations of the apo-5HT₃R-Salipro model, while a continuous water channel was observed for the serotonin-5HT₃R-Salipro model (Fig. 4a). Interestingly, no wetting was observed for the serotonin-5HT₃R-detergent model (Fig. 4a), during our unbiased, unrestrained simulations. In previously reported simulations of this structure where water permeation was observed, restraints were applied on the backbone atoms to preserve the resolved conformation[18]. Thus, while the pore in our open-state structure remained open and permeable over the course of the simulations without the application of restraints, the pore of the serotonin-5HT₃R-detergent structure did not (Fig. 4a, b and Supplementary Fig. 6c). Additionally, sodium permeation events were observed during the simulations of the serotonin-5HT₃R-Salipro form (Fig. 4c and Supplementary Movie 2). Ions

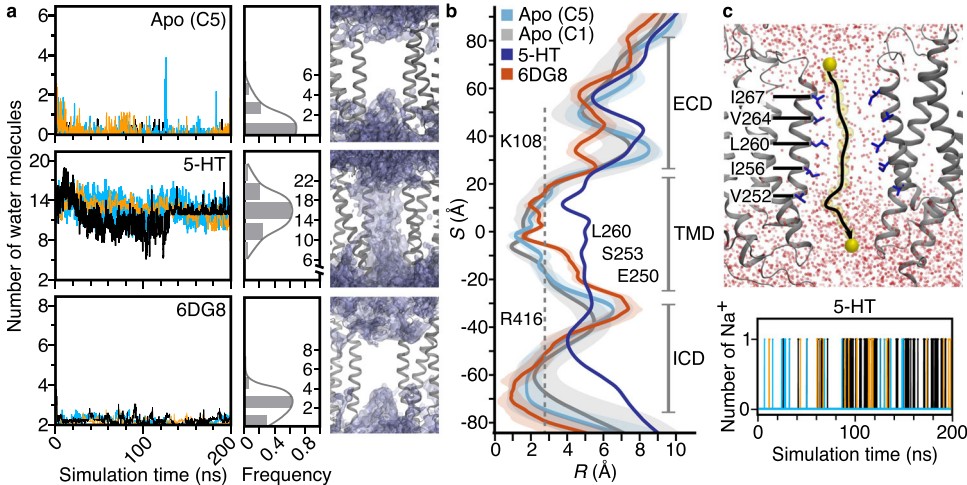

**Fig. 4 Pore permeability. a** Averaged water molecule density within 4 Å of L260 at the TMD. Left panel—number of water molecules, solid lines in different colours indicate results of three simulations and the histograms show the averaged distribution of the number of observed water molecules. Right panel—water molecule densities are shown in purple, two subunits of each model shown in grey. **b** Pore radius (R) profile relative to the distance along the receptor pore axis (S) derived from the MD simulations; solid lines represent the average radii over the last 50 ns of the simulations and shading the corresponding standard deviation. The dashed grey line represents the radius of a hydrated sodium ion (2.76 Å). **c** Top panel—the trajectory (black line) of a single sodium ion (yellow) translocated across the pore (Supplementary Movie 2). Bottom panel—the number of sodium ions translocated throughout three 200-ns simulations.

passed through the M2 region virtually unhindered within approximately 2–3 ns, suggesting that the conformation and flexibility of M2 observed in the 5HT₃R-Salipro form may be essential for channel permeation and that the lipid environment supports a conductive open state. Despite the channel adopting an open pore, the naturally markedly low (sub-pS) conductance of the receptor[30], means that permeation events are rare; we only observed four permeation events during our simulations, precluding robust assessment of conductivity. Nevertheless, the observed minimal TMD pore diameter of 9.4 Å (at residue I267) for our static structure (Supplementary Fig. 1e), and 7.2 ± 1.6 Å during the MD simulations (Fig. 4b and Supplementary Fig. 6c), is in very good agreement with previous reports of minimal channel diameters of 7.6–10.7 Å based on permeability data for organic cations for 5HT₃R expressed in mammalian cells[31,32]. Thus, the wide pore observed for the open-state serotonin-bound 5HT₃R in lipid bilayers is consistent with previous functional studies of the receptor. It should be noted that pore profiles are influenced by resolution and rotameric state of the pore-lining residues[17], explaining the difference between the static pore diameter and that determined by MD.

**Lipid-5HT₃R interactions facilitate a compact 'coupled' conformation.** In our cryo-EM maps, the TMD is surrounded by additional, mostly disordered density corresponding to the saposin-lipid belt (Fig. 1a and Supplementary Fig. 1f). Nevertheless, some more ordered moieties were observed at the outer leaflet adjacent to M4 in all maps, and between subunits in the serotonin-5HT₃R-Salipro map only (Fig. 5a). The shape and size of the former are suggestive of cholesterol (Fig. 5b). Indeed, M4 is a well-documented sterol binding site in nAChR[13,33]. Cholesterol occupied similar positions in the apo- and serotonin-5HT₃R-Salipro structures, nested in a hydrophobic pocket between pre-M1, M1, post-M4, and the Cys loop (Fig. 5b, c). In the apo-5HT₃R-detergent form the distance between post-M4 and the TM bundle is markedly larger than for the receptor in Salipro (Fig. 5d), precluding the stabilisation of (1) pre-M1 through a cation–π interaction between R217 and W459 observed in the presence of cholesterol, and (2) the Cys loop through the

interactions between F144 and the hydrophobic pocket. A similar picture is seen for the serotonin-bound state in detergent compared to Salipro (Fig. 5e). Overall, this cholesterol-mediated packing affects the positioning of the Cys loop, pre-M1, and consequently the M2–M3 linker at the TMD-ECD interface; in both 5HT₃R-Salipro conformations, a network of interactions involving a cluster of highly conserved residues[34] encompassing R218 (pre-M1), W187 (β9), F142, and D145 (Cys loop) is found adjacent to the highly conserved proline residue (P274) in the M2–M3 linker (Fig. 5f, g), which has a crucial role in receptor gating[1,2,35]. P274 is cradled between this cluster and the β1–β2 loop, whose position is stabilised by a salt-bridge between R218 and the likewise conserved E53 (β1–β2 loop). Notably, in the apo- and serotonin-5HT₃R-detergent form, residue R218 adopts a different conformation, possibly due to the lack of cholesterol stabilising the position of pre-M1. Consequently, the packing of the conserved cluster of residues is much looser and the salt-bridge between R218 and E53 is absent, resulting in a different conformation of the M2–M3 linker compared to the Salipro-reconstituted receptor. Thus, our structures suggest that the presence of cholesterol allosterically modulates the functioning of the channel through the stabilisation of M4, pre-M1, and the Cys loop in a tightly packed, 'M2–M3-linker-coupled' conformation, offering a structural basis for the 'uncoupled' state that is refractory to activation by ligand reported for the related nAChR in the absence of cholesterol[7]. Notably, 5HT₃R has been shown to reside in cholesterol-rich membrane domains[36] and cholesterol depletion also leads to diminished serotonin-induced currents in cells expressing 5HT₃R[8]. Interestingly, an open-state structure of 5HT₃R in detergent micelles to which cholesteryl hemisuccinate (CHS) and phospholipids were added prior to grid preparation (PDB accession code 6HIN)[17] was more similar to our open-state structure than the 6DG8 structure in some aspects (Fig. 6). Although no clear lipid densities were resolved, the structure showed a thinner TMD than the 6DG8 structure (43 Å, compared to 45 and 40 Å for the 6DG8 and serotonin-5HT₃R-Salipro structures, respectively), the distance between P220 (M1) and W246 (M4) was shorter (Fig. 6a; 10.7 Å, compared to 11.9 and 9.3 Å), and the R218-E53 salt-bridge seen in our structure was also present in the 6HIN structure (Fig. 6b). However, significant

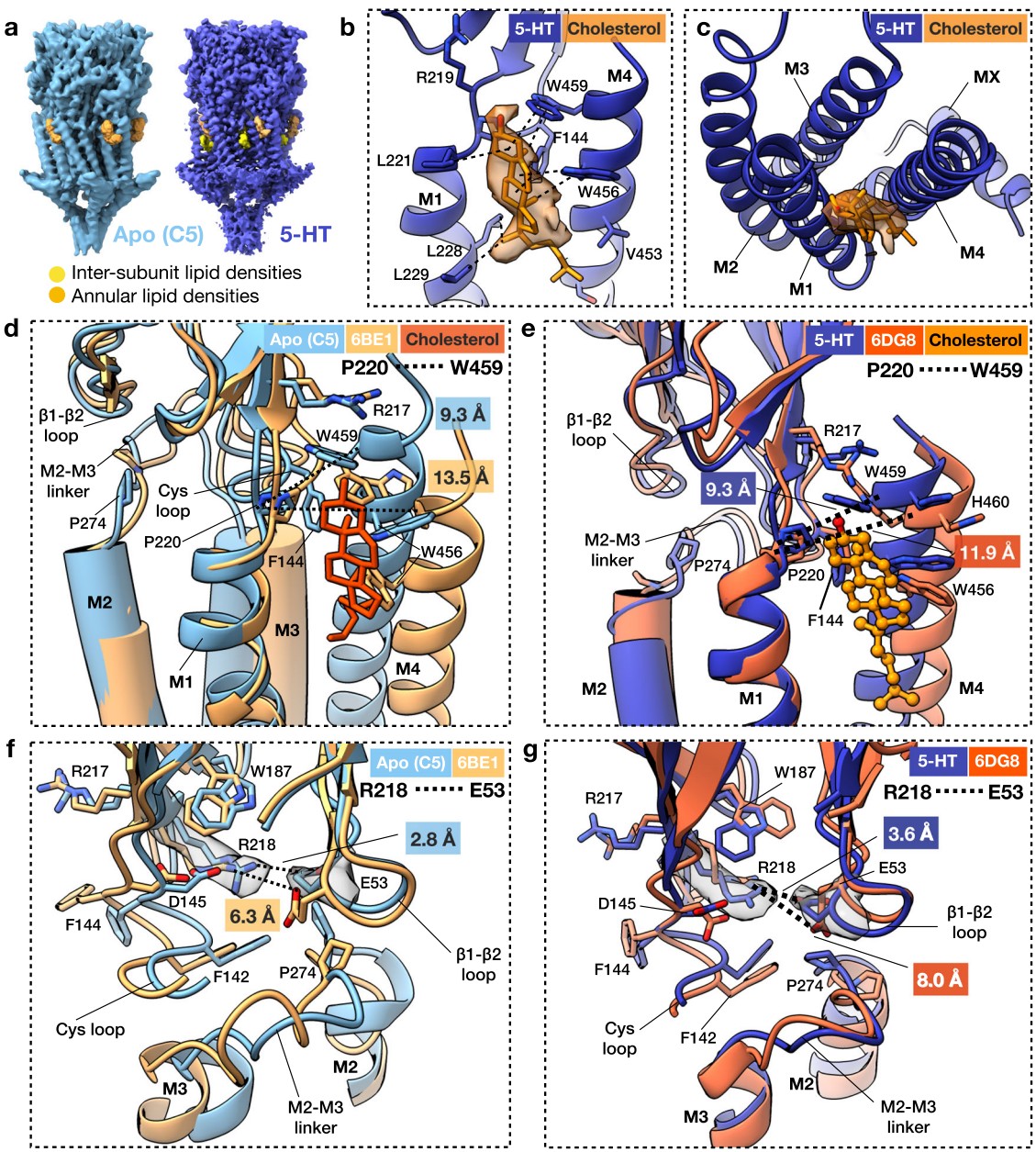

**Fig. 5 Lipid modulation of 5HT₃R. a** Additional densities corresponding to annular (orange) and inter-subunit lipids (yellow). **b** Tentative modelling of cholesterol at annular lipid site with density shown in transparent orange (contoured at σ 2.0). Putative lipid–protein interactions are indicated with dashed lines. **c** Top view of the putative cholesterol density. **d**, **f** Superposition of apo-5HT₃R-Salipro and apo-5HT₃R-detergent (6BE1, chain C) **d** at the cholesterol-binding pocket viewed from the membrane or **f** at the Cys loop, β1-β2 loop, M2–M3 linker junction viewed from the neighbouring subunit. Part of the Cys loop (C135-D138) and the TMD are omitted for clarity. Distances are labelled in the same colour scheme as the structures. Density sharpened using RELION with auto-determined b-factor is shown in transparent grey for residues R218 and E53 (contoured at σ 6.0). **e**, **g** Same as **d**, **f** but showing a superposition of the serotonin-bound 5HT₃R-Salipro (chain C) and 5HT₃R-detergent (6DG8) structures. Density for residues R218 and E53 is contoured at σ 6.5.

differences were observed between the TMDs of the 6HIN and serotonin-5HT₃R-Salipro structures (Fig. 6c), resulting in a smaller open pore (minimal diameter 6 Å) that would however still be wide enough to permit the passage of hydrated sodium ions (Supplementary Fig. 1e). Pore wetting and ion permeation of the 6HIN structure was indeed observed during previous MD simulations[17]. It should be noted that the simulated structure lacked the MA and MX helices as the local resolution of the ICD in the EM map precluded model building. The absence of the MA helices that normally line the lateral ion channel exit portals is expected to significantly affect simulations of ion permeation,

given that three Arg residues (R416, R420, and R426) residing on the MA helices are responsible for the aforementioned markedly low conductance of the 5HT₃ₐR[30,31]. A chimera of 5HT₃ₐR and 5HT₃ᵦR lacking these residues shows a 28-fold increase in single-channel conductance[30]. These considerations complicate direct comparisons of ion channel permeation based on MD simulations of the 6HIN and serotonin-5HT₃R-Salipro structures. It is further worth noting that the 6DG8 and 6HIN structures of the detergent-solubilised receptor also differ substantially from each other in the TMD (Fig. 6d). Taken together, these observations suggest that the TMD of the serotonin-bound 5HT₃R can adopt

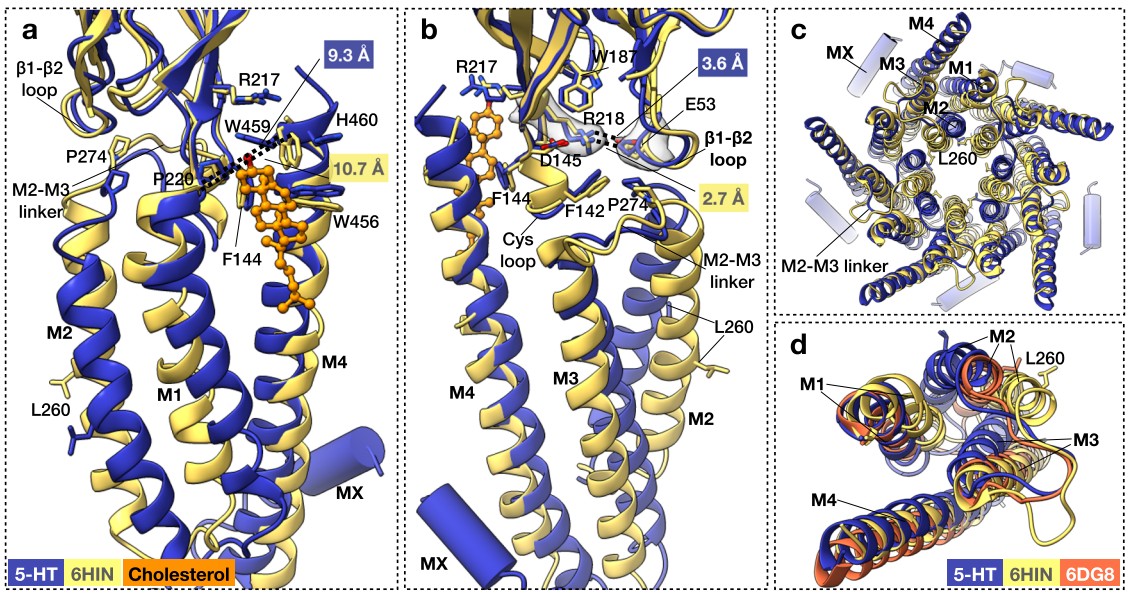

**Fig. 6 Comparison of serotonin-bound 5HT₃R-Salipro and 6HIN structures. a, b** Superposition of the serotonin-bound 5HT₃R-Salipro (chain C) and 5HT₃R-detergent (6HIN)[17] structures at **a** the cholesterol-binding pocket viewed from the membrane (M3 is omitted for clarity) or **b** at the Cys loop, β1-β2 loop, M2–M3 linker junction viewed from the neighbouring subunit (M1 and part of the Cys loop and TMD are omitted for clarity). Distances are labelled in the same colour scheme as the structures. Density sharpened using RELION with auto-determined b-factor, shown in transparent grey for residues R218 and E53, at σ 6.5. **c, d** Top views of the TMD, with **d** including the 5HT₃R-detergent 6DG8 structure and showing single subunits only for clarity.

different conformations depending on its lipid or (mixed) detergent micelle environment, with the presence of lipid promoting a more compact packing of the M1–M4 helices. Given that the serotonin-5HT₃R-Salipro form showed the widest pore with a diameter in accordance with previous functional studies[31,32], the presence of a lipid bilayer may be required to fully stabilise the maximally open state.

**Inter-subunit lipid densities**. Five additional densities observed at the inter-subunit allosteric binding pockets in the serotonin-5HT₃R-Salipro map resembled partially ordered phospholipids (Fig. 7a). Fitting of phosphatidylcholine (PC), PS, and PE lipid species (Fig. 7b), which were identified as significant components of the 5HT₃R-Salipro lipid bilayer (Supplementary Table 1), suggests that the density is most consistent with a lipid species with a small headgroup like PE or PS. Unsaturated PE (and to a lesser extent PA) lipids were previously shown to support receptor-gating in nAChR[5], suggesting that a small headgroup capable of forming intramolecular hydrogen bonds might indeed be favourable for native pLGIC function. Tentative modelling of POPE suggests the fatty acids of the interfacial lipid species chains make hydrophobic contacts with M1, M2, and M3, and that the phosphate moiety and headgroup make additional contacts with the M2–M3 linker (Fig. 7b). These inter-subunit lipid-like densities were absent in both the apo-C5 and apo-C1 maps, where the cavities between subunits were on average smaller than for the serotonin-bound 5HT₃R-Salipro conformation (177, 40–428, and 210–1198 Å³, respectively, see Supplementary Table 4). Another smaller, weaker additional density was observed in the largest inter-subunit cavity in the asymmetric apo-5HT₃R-Salipro map, which could potentially correspond to a fatty acid chain (Fig. 7a). These findings suggest that the lipid can only fully enter the inter-subunit cavity upon conformational changes in the TMD in response to gating, stabilising the open-pore conformation. Indeed, the inter-subunit TMD cavity is a well-documented site for allosteric modulation of pLGICs[37], and inter-subunit lipid binding was shown to lead to expansion of the pore and potentiation of agonist binding in the glutamate-gated chloride channel

GluCl[11]. Penetration of the TMD by lipids was also observed in a pre-activated state of 5HT₃R in a recent in silico study[19]. Although the main site of lipid intercalation differed, it was only observed for subunits undergoing conformational changes due to ligand binding, and the authors concluded that lipids support the activation process, in line with our observations.

**The asymmetric apo state might represent an activation intermediate**. To investigate the role of the consensus asymmetric apo state (apo-C1) in the activation process of the 5HT₃R, we compared it to the symmetric apo-C5 model and the consensus serotonin-bound conformation. For the ECD domain, no significant differences were observed between the five subunits of the apo-C1 form (RMSD 0.4–0.5 Å, Supplementary Table 2) or between each subunit and the apo-C5 form (RMSD 0.5–0.6 Å). The TMD-ECD interface was likewise found in a similar 'M2–M3-linker-coupled' cholesterol-stabilised conformation (Fig. 8a). Larger differences were observed between subunits at the TMD (RMSD 2.0–4.6 Å, Supplementary Table 2). Taking a cross-section at L260 (Fig. 8b), the carbon backbone displacement varied from 1.8 to 3.8 Å towards (chain A/C/D) or away from (B/ E) the pore axis (Supplementary Table 3), resulting in an overall slightly more constrained pore compared to the symmetric apo-C5 form (Figs. 1b and 8b). While some of the apo-C1 subunits are shifted closer to their counterparts in the serotonin-bound receptor relative to their positions in the apo-C5 form (e.g. chain A), the opposite is true for other subunits (e.g. chain D), and overall similar rotations and translations of the individual subunits are required to transition to the open-state structure (Fig. 8b and Supplementary Table 3). Thus, based on the comparison of the static structures alone, the position of the asymmetric apo state in the activation sequence of the receptor relative to the symmetric apo state is unclear. However, MD simulations of all 5HT₃R-Salipro models suggest that the apo-C1 model represents an intermediate conformation between the apo-C5 and the serotonin-5HT₃R-Salipro model. Firstly, its M2 helices were more flexible than those of the apo-C5 state but less flexible than those of the serotonin-bound state (Supplementary Fig. 7c),

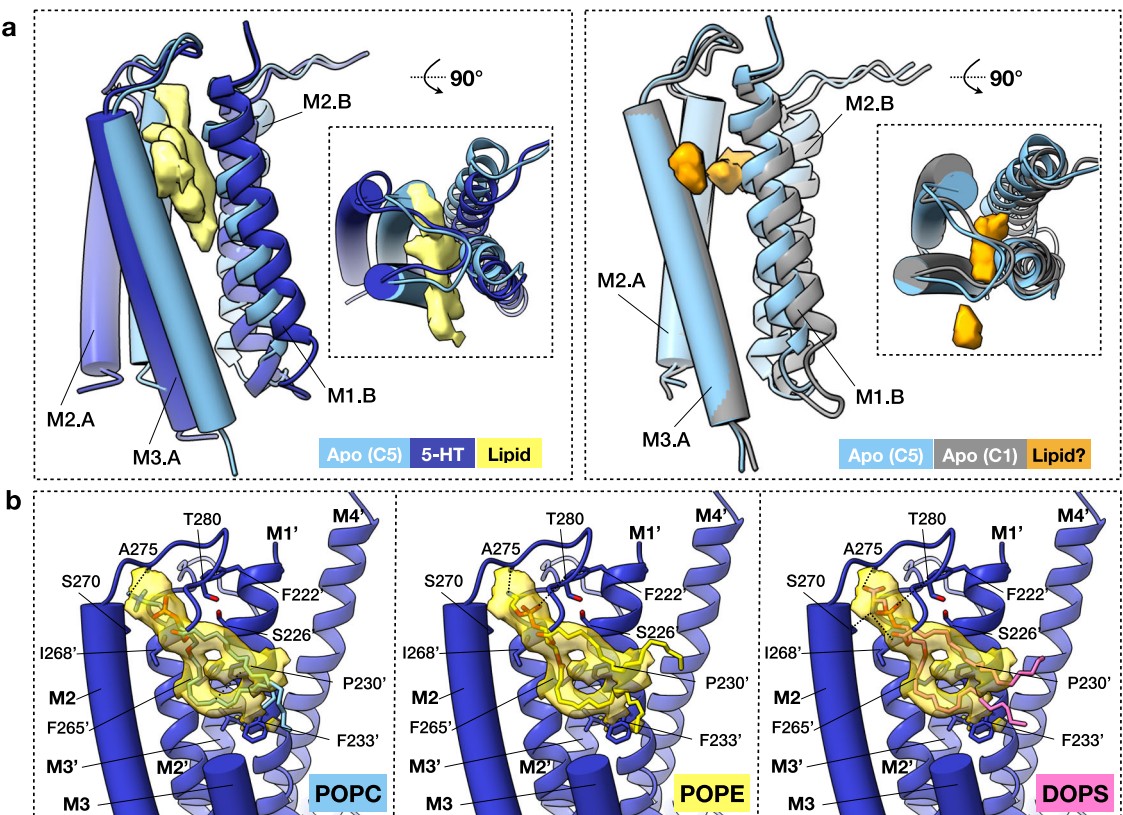

**Fig. 7 Inter-subunit lipid-like EM densities. a** Additional densities tentatively assigned as lipids (yellow) were seen in between subunits at the TMD for the serotonin-5HT₃R-Salipro EM map (left). Weaker densities were seen in the largest inter-subunit cavity of the apo-C1 EM map (right). **b** Tentative modelling of POPC (left), POPE (middle), and DOPS (right) in the inter-subunit phospholipid-like density (transparent dark yellow, contoured at σ 2.0) of the serotonin-5HT₃R-Salipro EM map. Residues P281-L293 of the M3 helix (of chain A) are omitted for clarity. TMD helices of chain A and B are shown in tube and ribbon representation, respectively.

bending at the intracellular end as observed for the serotonin-bound receptor, albeit less strongly (Supplementary Fig. 7a, b). Secondly, the apo-C1 state showed a slightly more hydrated channel than the apo-C5 state (Figs. 4a and 8c). Thirdly, to investigate the capacity for channel opening of both apo-5HT₃R-Salipro forms, five serotonin molecules were placed in the LBPs of each model, and triplicate 200-ns simulations were performed. Analogous simulations were performed for the apo structure crystallised in presence of detergent (PDB 4PIR)[15]. Although it has to be noted, that to observe the full transition to an activated state upon ligand docking much longer simulation times would be required[19], interestingly, wetting of the pore was observed for both apo-5HT₃R-Salipro forms, but not for the 4PIR structure, further supporting the allosteric role of the lipid environment in stabilising receptor conformations that are primed for gating (Fig. 8c). Furthermore, ligand docking yielded a slightly more hydrated channel for the apo-C1 state than for the apo-C5 state (Fig. 8c), suggesting the former may represent an intermediate, more readily activated state.

## Discussion
Based on the data presented here and a previous functional study[29], we propose a hypothetical model for allosterically lipid-modulated asymmetric activation of 5HT₃R upon serotonin binding (Fig. 9). Considering the highly similar conformation of the ECDs of the different subunits, there is no clear chemical determinant for the asymmetry observed in the TMD of the fully ligand-bound serotonin-5HT₃R-Salipro form. Various degrees of mean channel open times have been observed in response

to different numbers of consecutive/non-consecutive ligands in single-channel recordings of an α7-nAChR/5HT₃R chimera: one or two ligands at consecutive sites results in brief stable currents; two ligands at non-consecutive sites results in channel currents with intermediate mean open times; and three ligands at non-consecutive sites produce maximal mean open times[29]. This suggests that particular functional states could be ascribed to asymmetric stimuli. Thus, we speculate that asymmetric opening may result from sequential ligand binding yielding sequential conformational changes stabilised by the introduction of inter-subunit lipids during the transition of each subunit from a resting to an activated conformation. Such a mechanism would result in a large degree of conformational plasticity as we indeed observed (Fig. 3). Although no intermediate (closed) states with partial ligand occupancy were resolved, since the serotonin-5HT₃R-Salipro structure was obtained using saturating ligand concentrations, intermediate sub-stoichiometrically liganded states are expected based on the aforementioned single-channel recordings[29]. A recent in silico study of 5HT₃R reported asymmetric closed conformations which were attributed to sub-stoichiometric ligand occupation and stabilised by lipids penetrating into the TMD as discussed above[19], lending further support to our hypothesis. Notably, several closed intermediate states have been postulated to describe the single-channel activation of acetylcholine and serotonin receptors[38–41]. In the ligand-free resting state, the serotonin receptor is dynamic and displays a conformational equilibrium between symmetric and asymmetric forms. Deviations from the symmetry of the apo receptor might also be stabilised through interactions with the lipid bilayer. Our MD simulations suggest that there may be

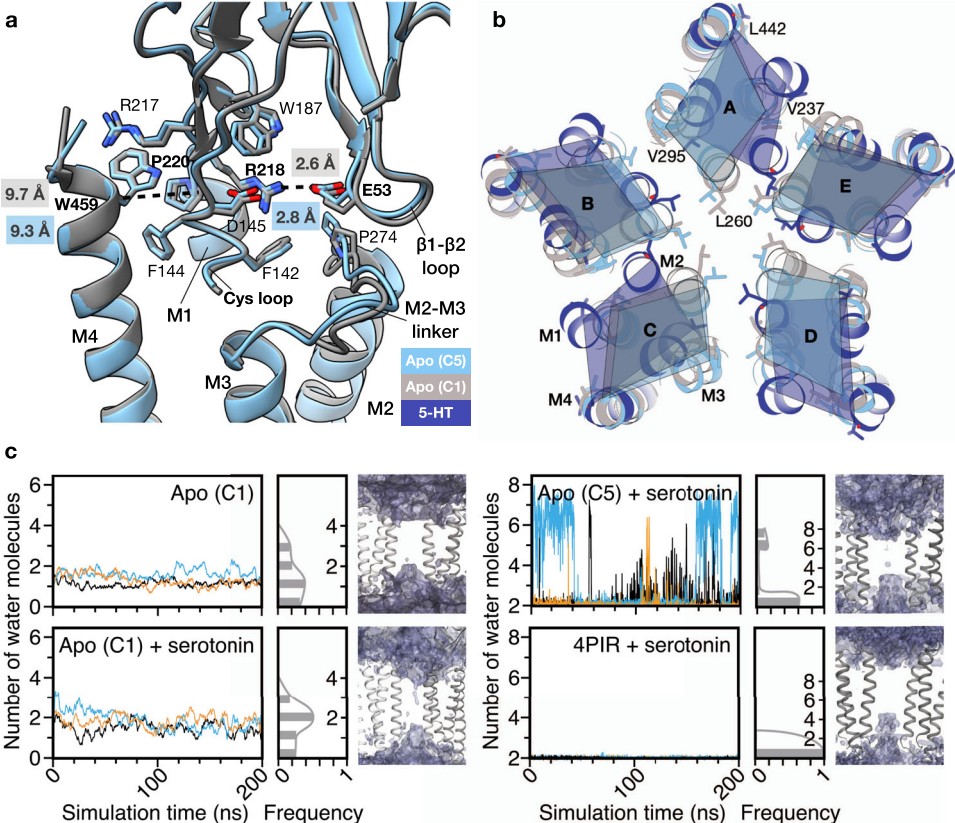

**Fig. 8 Comparison of symmetric and asymmetric apo conformations. a** Superposition of apo-C1 (chain C, grey) and apo-C5 5HT$_3$R-Salipro (light blue) structures. Part of M1 and the Cys loop (C135-Y140) are omitted for clarity. The Cα–Cα distance between P220 (pre-M1) and W459 (M4) ranged between 9.6 and 9.9 Å for the individual chains. The Cα–Cα distance between D53 (β1-β2 loop) and R218 (pre-M1) ranged between 2.5 and 3.3 Å. **b** Cross-section of the TMD at L260 (9′ position) of M2 for apo-C5, apo-C1, and serotonin-bound (dark blue) 5HT$_3$R-Salipro models. Displacement measured at the Cα atoms of residues on each helix in the same cross-section can be found in Supplementary Table 3. Superposition of the models that minimises the summed displacement of the indicated residues is shown. **c** Graphs show a number of water molecules within 4 Å of L260 at the TMD during MD simulations of apo-C1, and of apo-C1, apo-C5 (in a BPL-mimicking lipid environment) and 4PIR (in POPC) with five serotonin molecules docked into the ligand-binding pockets (+serotonin). Differently coloured lines represent the results of three simulations. Histograms show the averaged distribution of observed water molecules. Figures show water molecule densities in purple, with two subunits of each model shown in grey.

differences in the energy barrier to transition from the apo-C1 or apo-C5 state to the activated state (Fig. 8c), and it has been postulated that unbalanced forces promote state transitions[42]. Thus, we speculate that the less thermodynamically favourable asymmetric form may represent an intermediate state in the gating cycle.

The existence of asymmetric structural forms has also been reported for other pLGICs[10,43,44]. For example, an asymmetric closed conformation was solved by cryo-EM for the detergent-solubilised human α1β2γ2 GABA$_A$R in the presence of its native agonist GABA and an antagonist[10]. Interestingly, the authors also observed a 60/40 population distribution between the resolved (pseudo)symmetric and asymmetric conformation. Both structures were stabilised by inter-subunit CHS moieties, with additional sterols intercalating into the widened α1β2 interface in the asymmetric structure. However, the asymmetric state resolved for GABA$_A$R deviates more substantially from C5 symmetry than our asymmetric apo-5HT$_3$R-Salipro form; the γ2-subunit was observed to collapse into the pore, blocking the permeation pathway. Furthermore, the authors were cautious to interpret the structures in terms of their physiological relevance, in part due to the presence of detergent, and it should be noted that such asymmetric conformations were not observed in later studies of the related α1β3γ2 GABA$_A$R in nanodiscs[9,26]. In addition, asymmetric states of the related prokaryotic homopentamer cation channel GLIC in lipid bilayers were

observed by high-speed atomic force microscopy[44], and in simulations[43]. Notably, activation of the unrelated prokaryotic homopentameric magnesium transport protein CorA has also been shown to be asymmetric[45]. Thus, intrinsically symmetric homopentameric channels can form asymmetric structures, and loss of symmetry may be a more general feature of ion channel activation.

In summary, the structures of lipid-embedded 5HT$_3$R presented here show a number of features pertinent to the transition of the receptor to an open active state. Substantial structural differences observed compared to previous detergent-based structures could be related to the effect of stabilising lipid moieties—in particular, a cholesterol molecule adjacent to M4; the receptor adopts a more tightly packed 'coupled' conformation which extends to a cluster of highly conserved residues at the ECD-TMD interface important for gating. Furthermore, the receptor TMD is considerably compressed along the central axis due to the thickness of the hydrophobic core of the lipid bilayer. In consequence, the TMD helices are tilted, yielding a widened open channel. Thus, the 5HT$_3$R appears to be able to adopt various conformations depending on its specific environment[16–18], and our work provides a structural basis for allosteric modulation of 5HT$_3$R gating by the lipid membrane, which may extend to other pLGICs. Future developments in the field of in situ structural biology will be required to shed more light on the structure of the receptor in its native environment.

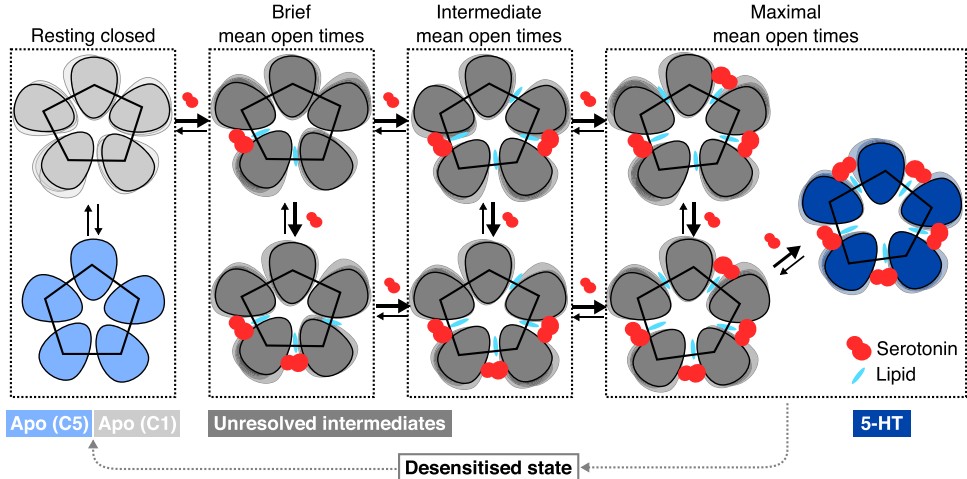

**Fig. 9 Model for asymmetric activation of 5HT₃R.** Proposed model for allosterically lipid-modulated asymmetric activation of 5HT₃R upon serotonin binding, based on 5HT₃R-Salipro structures, MD, and previous observations from single-channel recordings[29]. Various degrees of mean channel open times have been observed in response to different numbers of consecutive/non-consecutive ligands bound[29]. We speculate that the asymmetric apo form represents an intermediate state; that asymmetric opening may result from sequential ligand binding (5-HT shown in red) via this state yielding sequential conformational changes stabilised by the introduction of inter-subunit lipids (shown in light blue) during the transition of each subunit from a resting to an activated conformation; and that, considering that the channel already shows maximal mean open times upon binding three non-consecutive ligands[29], one ligand-binding event results in the introduction of two inter-subunit lipids as depicted (adjacent to the principal and the complementary ligand-binding subunits), thus resulting in stabilisation of the open state by five lipids upon binding of three non-consecutive ligands. Light blue, light grey and dark blue pentamers correspond to the resolved apo-C5, apo-C1 and serotonin-bound 5HT₃R-Salipro conformations, and dark grey pentamers to various unresolved asymmetric conformations with partial ligand occupancy.

In addition, the serotonin-bound receptor in lipid discs shows asymmetric opening, which we hypothesise to occur via an intermediate asymmetric apo state. Compared to the symmetric apo state or a previous symmetric state solved in detergent, the asymmetric apo and ligand-bound states display higher conformational flexibility in the pore-lining M2 helices, which may be important for efficient ion permeation. The observation of both symmetric and asymmetric 5HT₃R-Salipro conformations broadens our understanding of 5HT₃R gating. Nevertheless, the presented asymmetric activation model is speculative and what causes asymmetry in the homopentameric 5HT₃R remains an unanswered question. Future studies guided by the structural detail provided by our models will be necessary to address this point.

## Methods

**Production of saposin.** Saposin A fused to an N-terminal hepta-His tag separated by a TEV protease recognition site (sequence: MHHHHHHHENLYFQSSLPCDIC KDVVTAAGDMLKDNATEEEILVYLEKTCDWLPKPNMSASCKEIVDSYLPVIL DIIKGEMSRPGEVCSALNLCES) cloned into a pET-21 vector was expressed in Rosetta-gami 2(DE3) *E. coli* cells. Specifically, cells were grown in 5 L flasks at 37 °C, 180 rpm, in TB medium supplemented with 100 µg/mL ampicillin to high density (OD600 ≅ 2) at which point expression was induced by addition of 0.7 mM IPTG. The temperature was lowered to 30 °C, and cells were harvested 16 h post-induction. The cell pellet was flash-frozen in liquid nitrogen and stored at −80 °C until further use. Cells were resuspended in lysis buffer (20 mM HEPES, 150 mM NaCl, pH 7.5), supplemented with protease inhibitors, 1 mM MgCl₂, 1 mg DNAse I, and 1 mg/mL lysozyme, and subsequently disrupted using a Microfluidizer processor (Microfluidics M-110L) at 80 psi. The cell lysate was clarified by centrifugation (26,000 × g, 30 min), and heated at 85 °C for 10 min in a water bath. The debris was removed by centrifugation (26,000 × g, 30 min), and the supernatant was passed through a 0.45-µm-pore filter. Imidazole was added to the filtrate to a final concentration of 20 mM, and the sample was applied at a flow rate of 2 mL/min to a 5 mL HisTrap HP column (GE Healthcare) equilibrated with lysis buffer supplemented with 20 mM imidazole. The column was washed with 25 column volumes (CV) of lysis buffer supplemented with 20 mM imidazole, followed by 25 CV of lysis buffer supplemented with 40 mM imidazole. Saposin was eluted with 10 CV of lysis buffer supplemented with 500 mM imidazole. Peak fractions were pooled, TEV protease was added at a 1:1 molar ratio, and the sample was dialysed overnight against dialysis buffer (20 mM HEPES, 300 mM NaCl, pH 7.5) using SnakeSkin dialysis tubing, MWCO 3,500 (ThermoFisher Scientific). The

sample was filtered, and cleaved saposin was separated from TEV protease and any uncleaved material using a 1 mL HisTrap HP column (GE Healthcare) equilibrated with dialysis buffer. The flow-through was collected, followed by a 20 mM imidazole wash to remove non-specifically bound cleaved saposin, and a 20–500 mM imidazole gradient over 20 mL to recover any uncleaved saposin and TEV. Fractions were analysed by SDS-PAGE, and fractions corresponding to cleaved saposin were pooled. Initially, the sample was subsequently subjected to size-exclusion chromatography (SEC): the concentrated sample was filtered through a 0.22-µm-pore centrifugal filter and submitted to size-exclusion chromatography (SEC) on a Superdex 75 10/300 column (GE Healthcare) equilibrated with 20 mM HEPES, 150 mM NaCl, pH 7.5. As the sample consistently appeared homogeneous, this step was skipped in later purifications. The final sample was concentrated to 1–2 mM using an Amicon Ultra centrifugal concentrator (MWCO 3,500).

**Cell culturing, 5HT₃R expression, purification and reconstitution in saposin.** A stable T-Rex-293 cell line was used for the expression of a murine wild-type 5-HT₃A receptor construct containing four N-terminal StrepII tags[15]. Cells were adapted to serum-free suspension culture in FreeStyle-293 expression medium (ThermoFisher Scientific) and cultivated at 37 °C in 1 L flasks under orbital agitation at 120 rpm. Protein expression was induced by adding 4 µg/mL tetracycline when the cell density reached ~4 × 10⁶ cells/mL, at which point the temperature was lowered to 30 °C. Sodium butyrate was added to a final concentration of 6.8 mM 4–6 h post-induction. The typical total culture volume was 4 L. Cells were harvested after 48 h, flash-frozen in liquid nitrogen, and stored at −80 °C until further use. Cells were resuspended in 10 mM HEPES, 1 mM EDTA, pH 7.4, supplemented with protease inhibitors (2 µg/mL leupeptin, 2 µg/mL pepstatin A, 0.2 mg/mL benzamidine-HCl, 20 µg/mL AEBSF, and 3 µg/mL aprotinin) using a dounce homogeniser, and subsequently disrupted using a Microfluidizer processor (Microfluidics M-110L) at 80 psi. Cell debris was removed by centrifugation (10,000 × g, 45 min), and membranes were collected from the clarified lysate by ultracentrifugation for 6 h at ~130,000 × g. The membrane pellet was flash-frozen in liquid nitrogen and stored at −80 °C until further use. The membrane pellet was resuspended in 50 mM Tris, 500 mM NaCl, pH 8 (final concentrations). C12E9 (Anatrace) was added dropwise to a final concentration of 0.75% (w/v), and the membranes were solubilised under gentle stirring for 2 h. Non-solubilised material was removed by centrifugation for 1 h at 49,000 g. The supernatant was then passed through a 0.45-µm-pore filter and applied to a 5 mL Streptactin Superflow high capacity column (IBA), equilibrated with 50 mM Tris, 150 mM NaCl, 0.01% C12E9, pH 8 (Strep buffer), at a flow rate of 1 mL/min. The column was washed with 250 mL of Strep buffer, and the receptor was subsequently eluted with 10 mM D-desthiobiotin in Strep buffer. The peak fractions were pooled and concentrated to 300–500 µL (~10–30 µM) using an Amicon Ultra centrifugal concentrator (MWCO 100,000, Merck). For tests with the sample in detergent, the sample was then filtered through a 0.22 µm-pore centrifugal filter and submitted to SEC on a Superose 6 Increase 10/300 column (GE Healthcare) equilibrated with 25 mM

HEPES, 125 mM NaCl, pH 7.4 (SEC buffer) supplemented with 0.01% C12E9. Alternatively, the purified receptor was reconstituted using saposin into porcine brain polar lipid (BPL, Avanti Polara Lipids) 5HT₃R-Salipro particles. Specifically, BPL was resuspended at 83 mg/mL in 25 mM HEPES, 150 mM NaCl, 5% C12E9, pH 7.4, sonicated 3 × 1 min in a bath sonicator followed by ten freeze/thaw cycles, and incubation at room temperature for a minimum of 4 h on an orbital rocker. Purified 5HT₃R was incubated for 15 min at room temperature on an orbital rocker with a 450-fold molar excess of detergent-destabilised BPL, assuming a molecular weight of 650 g/mol for BPL. Saposin was then added to 30-fold molar excess relative to 5HT₃R, and the sample was incubated for 15 min at room temperature. Subsequently, the sample was incubated for 15 min at room temperature with 100–200 mg of wet BioBeads SM2 resin (Bio-Rad), and this last step was repeated once. The sample was then filtered through a 0.22 μm-pore centrifugal filter and submitted to SEC on a Superose 6 Increase 10/300 column (GE Healthcare) equilibrated with SEC buffer. SEC peak fractions were pooled and concentrated to ~0.5–2.5 mg/mL.

**EM grid preparation and cryo-EM single-particle application.** The different SEC peak fractions of 5HT₃R-Salipro were checked by negative staining using a Tecnai Spirit Biotwin at 120 kV (ThermoFisher Scientific). The most homogenous fractions were pooled and concentrated to around 0.43 mg/mL. Grids were prepared in the presence and absence of ligand (serotonin-5HT₃R-Salipro and apo-5HT₃R-Salipro, respectively). For the serotonin-5HT₃R-Salipro sample, the receptor was mixed with 100 μM serotonin and 2 mM CaCl₂, and incubated for 45–60 min on ice. The addition of CaCl₂ is known to promote fast desensitisation and recovery from desensitisation[46,47]. For both samples 0.125% (w/v) CHAPS (0.25xCMC) was added right before freezing. 3 μl of the sample was then applied onto a plasma-cleaned (NanoClean, model 1070, Fischione Instruments, with Ar/O₂ atmosphere, a ratio of 95/5, at 35 W for 35 s) R2/2 300 mesh Au-carbon (for serotonin-5HT₃R-Salipro) and UltraAuFoil (for apo-5HT₃R-Salipro) grids (Quantifoil). Grids were blotted at 100% humidity and 4 °C and then plunge-frozen in liquid ethane using a Mark IV Vitrobot. The grids prepared with and without the addition of CHAPS before freezing were screened; final data sets were collected from grids prepared with CHAPS, which showed better particle orientation distribution and thinner ice. Data sets of both apo- and serotonin-5HT₃R-Salipro samples were collected on a Titan Krios electron microscope with a Falcon III camera (ThermoFisher Scientific), using a pixel size of 0.832 Å. For the apo-5HT₃R-Salipro sample, two data sets were collected on different dates from the same grid using the same imaging settings. Further data collection details are given in Supplementary Table 5.

**Image processing.** During data collection, all data sets were pre-processed 'on-the-fly' using Warp[48] for frame alignment, contrast transfer function (CTF) estimation, and particle picking, and cryoSPARC[49] was used for 2D classification to monitor data quality including particle orientation and integrity. Warp was used to exclude suboptimal micrographs (25 for apo-5HT₃R-Salipro, none for serotonin-5HT₃R-Salipro). Subsequently, data were imported into RELION 3.0[50,51]. Motion correction and dose-weighting were performed with MotionCor2[52] using a 5 × 5 patch. CTF estimation was performed with CTFFIND 4.1[53] for the apo-5HT₃R-Salipro data set and Gctf-v1.06[54] for the serotonin-5HT₃R-Salipro data set. Autopicking was performed using crYOLO[55] for the apo-5HT₃R-Salipro data set and Warp for the serotonin-5HT₃R-Salipro data set. Particles were extracted with a box size of 320 pixels (266 Å) and imported into cryoSPARC for 2D classification, 3D heterogeneous refinement (classification with C5 symmetry imposed) and 3D homogeneous refinement. The 3D classification was attempted both with and without applying symmetry, with the former resulting in better separation of the particles. A star file of the selected good particles was created using UCSF Pyem[56] and re-imported into RELION 3.0. Particles were further 3D classified without alignment using C1 symmetry in RELION 3.0. Most of the particles fell into one class. For the apo-5HT₃R-Salipro data sets, after initial 3D classification and 3D refinement in cryoSPARC, subsequent 3D classification in RELION 3.0 without alignment and without imposing symmetry, yielded both symmetric and asymmetric maps. To probe the (a)symmetry of our final maps, particle sets were subjected to symmetry expansion along the C5 (pseudo)symmetry axis, and further 3D classification without alignment into 12 (for apo-5HT₃R-Salipro) or 16 (for serotonin-5HT₃R-Salipro) classes using a monomer mask. At the obtained resolution, the differences between the resulting classes for the apo-5HT₃R-Salipro particle set was negligible. The data sets were then subjected to several rounds of polishing, CTF refinement, and 3D refinement with C5 symmetry imposed (for apo-C5) or without symmetry imposed (apo-C1, and serotonin-5HT₃R-Salipro). Local resolution estimation was performed using RELION, and the directional resolution was estimated using 3DFSC[57].

**Model building and analysis.** The initial model was built manually in Coot[58] with PSIPRED 4.0[59] secondary structure prediction and the published murine serotonin receptor x-ray structure (PDB accession code 4PIR) for guidance. RELION 3.0 with automatic b-factor estimation was used to sharpen the maps. PHENIX[60] was used for model refinement, and the model was scored with Molprobity[61]. The refined model was then used to locally sharpen the refined EM map using LocScale[62]. The LocScale-sharpened EM map was subsequently used for further iterative

refinement in Coot and PHENIX. The models were then used to generate density maps from all their specified atoms using UCSF Chimera[63] at a resolution of 2 Å, for performing model-map cross-validation using RELION 3.0. At the ECD, resolved densities corresponding to the N-linked glycosylation adjacent to N82, N148 and N164 (Fig. 1a) were not modelled. Pore profiles of the models were determined using HOLE[64] and visualised in VMD[65]. CASTp[66] analysis was used to calculate the volume of the inter-subunit cavity. Structural figures were prepared using UCSF Chimera[63], PyMOL[67], VMD[65], GIMP and Inkscape.

**Molecular dynamics simulations.** All protein models were prepared using the Schrödinger software suite under the OPLS_2005 force field[68]. Hydrogen atoms were added to the repaired cryo-EM structures at physiological pH (7.4) with PROPKA[69] tool to optimise the hydrogen bond network provided by the Protein Preparation tool in Schrödinger. Constrained energy minimisations were carried out on the full-atomic models until the RMSD of the heavy atoms converged to 0.4 Å. All liganded receptor structures were prepared using the Schrödinger 2015 software suite; the LigPrep module was used for geometric optimisation applying the OPLS_2005 force field. The ionisation state of ligands was calculated with the Epik[70] tool employing the Hammett and Taft methods in combination with ionisation and tautomerisation tools[70]. The final pose of the docked ligands was similar to the resolved posed for the serotonin-bound structure.

Membrane systems were built using the membrane building tool CHARMM-GUI[71] and the OPM (Orientations of Proteins in Membranes) webserver (https://opm.phar.umich.edu/ppm_server)[72] was used to align the experimental structures in the lipid bilayer. To mimic the lipid composition of BPL, we used POPC (1-palmitoyl-2-oleoyl-glycero-3-phosphocholine), POPS (1-palmitoyl-2-oleoyl-sn-glycero-3-phospho-L-serine), POPE (1-palmitoyl-2-oleoyl-sn-glycero-3-phosphoethanolamine), and cholesterol at a molar ratio of 15:22:39:24. The simulation system contained a total of 300 lipids for simulations of the 5HT₃R-Salipro structures. For the simulations of the 6DG8 structure, we included 244 POPC lipids to mimic the simulation environment used in the original work. In addition, 0.15 M NaCl was added to each simulated system. We modelled the protein, lipids, water, and ions using the CHARMM36m force field[73,74]. Ligands were assigned with CHARMM CgenFF force field[75]. Ligand geometry was submitted to ORCA[76] for optimisation at the Hartree-Fock 6-31G* level when generating force field parameters. The system was gradually heated from 0 to 310 K followed by a 1 ns initial equilibration at constant volume with the temperature set to 310 K. The protein backbone and heavy atoms of the ligands were restrained during the equilibration steps with a force constant of 3 kcal/(mol Å²), after which restraints were released and a further 200 ns of simulation was run. Excluding the MX helices (residues 309–334) because of their high flexibility due to missing disordered linker regions, total backbone RMSD calculations indicated that the simulation systems stabilised within ~100 ns (Supplementary Fig. 6). Simulations were run in triplicate and the last 50 ns were used for analysis. All bond lengths to hydrogen atoms were constrained with M-SHAKE. Non-bonded interactions were treated using a force switch of 10–12 Å. Long-range electrostatic interactions were computed by the Particle Mesh Ewald (PME) summation scheme. All MD simulations were done in GROMACS[77]. The simulation parameter files were obtained from CHARMM-GUI website[71]. The radius profiles of M2 based on MD simulations were analysed using CHAP[78].

**Thermo-stability and microscale thermophoresis characterisation.** The thermal stability of the receptor in detergent and reconstituted into BPL Salipro particles, respectively, was determined using nanoDSF (NanoTemper Technologies). Samples obtained after SEC were diluted to 0.20–0.25 mg/mL and loaded into Prometheus NT.48 Series nanoDSF grade standard capillaries (NanoTemper Technologies). Samples were heated from 15 to 95 °C at 1 °C/min, and the ratio between intrinsic fluorescence at 330 and 350 nm after excitation at 280 nm was used to monitor receptor denaturation. The first derivative of the unfolding curves was used to determine the apparent melting temperature using PR.ThermControl software (NanoTemper Technologies).

The affinity of 5HT₃R-Salipro for serotonin was assessed by microscale thermophoresis (MST). 5HT₃R-Salipro was labelled with M-647-NHS dye (NanoTemper Technologies) as per the manufacturer's instructions. Labelled receptor (final concentration 9.5 nM) was added to a dilution series of 5-HT at 0.3 nM–11 μM (final concentrations) in 25 mM HEPES, 150 mM NaCl, pH 7.4, supplemented with 0.007% (w/v) Tween-20 (final concentration) using LoBind tubes (Eppendorf) to prevent adsorption of the sample. Under these conditions, the fluorescence was found to be constant (within ± 10%) for all points in the titration curve. The samples were loaded into premium coated capillaries (NanoTemper Technologies). MST experiments were carried out on a blue/red Monolith NT.115 (NanoTemper Technologies) using the red filter set, an excitation power of 100%, and an MST power of 60%. MST data were analysed in MO.Affinity Analysis software (NanoTemper Technologies), using −1 to 0 s before IR laser was turned on as the cold region, and 19 to 20 s after the IR laser was turned on as the hot region.

**Lipidomics.** Three biological replicates of 5HT₃R-Salipro (~1 nmol final) were prepared as described above. The gel filtration peak at ~17 mL corresponding to saposin discs without the receptor inserted (empty discs) were also collected for

each biological replicate and an additional fourth biological replicate. Lipids were extracted using a methyl-*tert*-butyl ether (MTBE) protocol[79]. Briefly, the samples were extracted in methanol/MTBE/water and the organic phase was dried, resuspended in 300 μL methanol:chloroform 1:1 and 12:0/13:0 PC, 17:0/20:4 PC, 14:1/17:0 PC, 21:0/22:6 PC (2 μM each), 17:1 LPC (1.5 μM), Cer/Sph-Mix LM6002 (1.5 μM), 12:0/13:0 PE, 17:0/20:4 PE, 14:1/17:0 PE, 21:0/22:6 PE (3 μM each), 12:0/13:0 PS, 17:0/20:4 PS, 14:1/17:0 PS, 21:0/22:6 PS (3 μM each), 12:0/13:0 PI, 17:0/20:4 PI, 14:1/17:0 PI and 21:0/22:6 PI (2 μM each) were added as internal standard (all lipid species are quantitative LM standards from Avanti Polar Lipids). All of the extracts were again dried and resuspended in 50 μL isopropanol: chloroform:methanol (90:5:5 v/v/v). Data acquisition was performed on a Q Exactive Focus Orbitrap instrument (ThermoFisher Scientific) coupled to a Vanquish UHPLC (ThermoFisher Scientific) according to previously published protocols[80,81]. Briefly, the chromatographic separation was performed on a BEH C8 column (100 × 1 mm, 1.7 μm, Waters, Milford, MA, USA) at 50 °C. Mobile phase A was deionised water containing 1 vol% of 1 M aqueous ammonium formate (final concentration 10 mM) and 0.1 vol% of formic acid as additives. Mobile Phase B was a mixture of acetonitrile/isopropanol 5:2 (v/v) with the same additives. Gradient elution started at 50% mobile phase B, rising to 100% B over 40 min; 100% B was held for 10 min and the column was re-equilibrated with 50% B for 8 min before the next injection. The flow rate was 150 μL/min, the samples were kept at 8 °C and the injection volume was 2 μL. The mass spectrometer was operated in Data Dependent Acquisition mode using a HESI II ion source. Every sample was measured once in positive polarity and once in negative polarity. Samples were measured in positive electrospray mode at 4.5 kV source voltage, 275 °C source temperature, and 300 °C capillary temperature in full scan (mass/ charge ratio ($m/z$), 160–1.150). In negative electrospray mode, the source voltage was at 3.8 kV, the source temperature was at 325 °C, and the capillary temperature was at 300 °C in full scan ($m/z$ 300 to 1.100). Full scan profile spectra were acquired in the Orbitrap mass analyser at a resolution setting of 70,000 at $m/z$ 200. For MS/MS experiments, the three most abundant ions and ions from the inclusion list (data-dependent fragmentation mode) of the full scan spectrum were sequentially fragmented. Data analysis was performed using Lipid Data Analyser, a custom developed software tool described in detail elsewhere[82,83]. Internal standards were added as a one-point calibration and normalised to total protein amount.

**Reporting summary**. Further information on research design is available in the Nature Research Reporting Summary linked to this article.

## Data availability

Data supporting the findings of this manuscript are available from the corresponding authors upon reasonable request. A reporting summary for this Article is available as a Supplementary Information file. Atomic coordinates of the murine 5HT₃R reconstituted into Salipro have been deposited at the Protein Data Bank (PDB) under accession codes PDB 6Y5A (serotonin-bound), PDB 6Y59 (apo-C5), and PDB 6Y5B (apo-C1). The cryo-EM maps have been deposited at the Electron Microscopy Data Bank under accession codes EMD-10692 (serotonin-bound), EMD-10691 (apo-C5), and EMD-10693 (apo-C1). The original movies and final particle data sets have been deposited at the Electron Microscopy Public Image Archive (EMPIAR), accession code EMPIAR-10381. Source data are provided with this paper.

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

## Acknowledgements

We thank Deryck Mills and the EM staff at Max Planck Institute of Biophysics for expert technical assistance; Juan Francisco Castillo Hernandez and Özkan Yildiz for computing support; Arne Möller for critical reading of the manuscript; and Luc Veya and Catarina Alves for input into sample preparation. Y.Z., P.M.D., R.M.S. and M.K. were supported by the Sofja Kovalevskaja Award to M.K. from the Alexander von Humboldt Foundation. Y.Z. was partially supported by IMPReS international student scholarship. R.M.S. was partially supported by a fellowship from SFB807 (funded by the German Research Fund, DFG). M.K. is further supported by the Heisenberg Programme from DFG (grant KU 3222/3-1). H.V. was supported by Chinese Academy of Science (Grant No. 2020FSB0003). S.Y. and R.Z. were supported by funding from Chinese Academy of Sciences, the Shenzhen Institutes of Advanced Technology, CAS, Shenzhen government (Grant No. JCYJ20200109114818703) as well as that from Guangdong province (Grant No. 2019QN01Y306). MD calculations were supported by the Interdisciplinary Centre for Mathematical and Computational Modelling in Warsaw (grant nos. GB70-3 & GB71-3).

## Author contributions

M.K. and H.V. conceived the project. P.M.D. and L.E.-S. performed cell culturing. P.M.D. optimised and performed protein preparation, saposin reconstitution, and biophysical analyses. Y.Z. performed the optimisation of EM grid preparation. Y.Z. and P.M.D. performed EM data collection, processing and model building. R.Z. and S.Y. performed the MD simulations and analysed the results. M.Z.-L. and H.K. performed lipidomics. R.M.S. contributed to the EM data analysis. Y.Z. and P.M.D. wrote the manuscript with input from H.V., S.Y. and M.K. M.K. supervised the project and provided the project funding. Y.Z. and P.M.D. share the first authorship.

## Funding

## Competing interests

H.V. and S.Y. are cofounders of AlphaMol Science Ltd. The remaining authors declare no competing interests.
