## [Peer Review File · Nature Communications]

Reviewers' Comments:

Reviewer #1:

Remarks to the Author:

The updated manuscript with the response of the RMSD is clearly the cause of the MX helix is good to see and I agree with the authors. This is acceptable for publication.

Reviewer #2:

Remarks to the Author:

The authors have added considerable text to the manuscript to present a more balanced perspective on the biological significance of their findings. The new structures are interesting and will certainly provoke further discussion and research. I support publication of the work in its current form, with the following minor modification:

In both the Abstract and Introduction, the authors note that their MD simulations suggest that their open conformation conducts ions while their MD simulations of a published open conformation does not. This is cherry picking data for reasons discussed in previous reviews. The authors should either perform MD simulations on both published open structures and report both findings, or focus the Abstract and Intro on the novel findings obtained in this study.

Please find our response to the comments of the reviewers (in bold).

Reviewer #1 (Remarks to the Author):

The updated manuscript with the response of the RMSD is clearly the cause of the MX helix is good to see and I agree with the authors. This is acceptable for publication.

Thank you very much.

Reviewer #2 (Remarks to the Author):

The authors have added considerable text to the manuscript to present a more balanced perspective on the biological significance of their findings. The new structures are interesting and will certainly provoke further discussion and research. I support publication of the work in its current form, with the following minor modification:

In both the Abstract and Introduction, the authors note that their MD simulations suggest that their open conformation conducts ions while their MD simulations of a published open conformation does not. This is cherry picking data for reasons discussed in previous reviews. The authors should either perform MD simulations on both published open structures and report both findings, or focus the Abstract and Intro on the novel findings obtained in this study.

We thank the reviewer for the suggestion and updated the abstract and the introduction in respect to the MD simulations of the previously published structures. The updated lined of the abstract reads as follows, the modifications are highlighted in yellow:

Line 33: ...We report three structures of the Cys-loop 5-HT_{3A} serotonin receptor (5HT_{3R}) reconstituted into saposin-based lipid bilayer discs: a symmetric and an asymmetric apo state, and an asymmetric agonist-bound **open** state. **In comparison to previously published 5HT_{3R} conformations in detergent**, the lipid bilayer stabilises the receptor in a **more** tightly packed, 'coupled' state, involving a cluster of highly conserved residues. In consequence, the agonist-bound receptor conformation adopts a wide-open pore capable of conducting sodium ions in unbiased molecular dynamics (MD) simulations **-which we did not observe for a previously published serotonin-bound, detergent-based 5HT_{3R} conformation**. Taken together, we provide a structural basis for the modulation of 5HT_{3R} by the membrane environment, and a model for asymmetric activation of the receptor."

In the introduction we removed the statement "**We did not observe full-pore wetting or sodium-ion conductance for a serotonin-bound detergent-solubilised structure under similarly unbiased MD simulation conditions**." On lines 89-91.